# DEEP GOAL-ORIENTED CLUSTERING

## ABSTRACT

Clustering and prediction are two primary tasks in the fields of unsupervised and supervised machine learning. Although much of the recent advances in machine learning have been centered around those two tasks, the interdependent, mutually beneficial relationship between them is rarely explored. One could reasonably expect appropriately clustering the data would aid the downstream prediction task and, conversely, a better prediction performance for the downstream task could potentially inform a more appropriate clustering strategy. In this work, we focus on the latter part of this mutually beneficial relationship. To this end, we introduce Deep Goal-Oriented Clustering (`DGC`), a probabilistic framework that clusters the data by jointly using supervision via *side-information* and unsupervised modeling of the inherent data structure in an end-to-end fashion. We show the effectiveness of our model on a range of datasets by achieving prediction accuracies comparable to the state-of-the-art, while, more importantly in our setting, simultaneously learning congruent clustering strategies. We also apply `DGC` to a real-world breast cancer dataset, and show that the discovered clusters carry clinical significance.

## 1 INTRODUCTION

Much of the advances in supervised learning in the past decade are due to the development of deep neural networks (DNN), a class of hierarchical function approximators that are capable of learning complex input-output relationships. Prime examples of such advances include image recognition (Krizhevsky et al., 2012), speech recognition (Nassif et al., 2019), and neural translation (Bahdanau et al., 2015). However, with the explosion of the size of modern datasets, it becomes increasingly unrealistic to manually annotate all available data for training. Hence, understanding inherent data structure through unsupervised clustering is of increasing importance.

Several approaches to apply DNNs to unsupervised clustering have been proposed in the past few years (Caron et al., 2018; Law et al., 2017; Xie et al., 2016; Shaham et al., 2018), centering around the concept that the input space in which traditional clustering algorithms operate is of importance. Hence, learning this space from data is desirable, in particular, for complex data. Despite the improvements these approaches have made on benchmark clustering datasets, the ill-defined, ambiguous nature of clustering still remains a challenge. Such ambiguity is particularly problematic in scientific discovery, sometimes requiring researchers to choose from different, but potentially equally meaningful clustering results when little information is available a priori (Ronan et al., 2016).

When facing such ambiguity, using direct side-information to reduce clustering ambivalence proves to be a fruitful direction (Xing et al., 2002; Khashabi et al., 2015; Jin et al., 2013). Direct side-information is usually available in terms of constraints, such as the *must-link* and the *cannot-link* constraints (Wang & Davidson, 2010; Wagstaff & Cardie, 2000), or via a pre-conceived notion of similarity (Xing et al., 2002). However, defining such direct side-information requires human expertise, which could be labor intensive and potentially vulnerable to labeling errors. On the contrary, indirect, but informative, side-information might exist in abundance, and may not require human expertise to obtain. Being able to learn from such indirect information to form a congruous clustering strategy is thus immensely valuable.

**Main Contributions** We propose *Deep Goal-Oriented Clustering* (`DGC`), a probabilistic model that is capable of using indirect, but informative, side-information to form a pertinent clustering strategy. Specifically: 1) We combine supervision via side-information and unsupervised data structure modeling in a probabilistic manner; 2) We make minimal assumptions on what form the

supervised side-information might take, and assume no explicit correspondence between the side-information and the clusters; 3) We train `DGC` end-to-end so that the model simultaneously learns from the available side-information while forming a desired clustering strategy.

## 2 RELATED WORK

Most related work in the literature can be classified into two categories: 1) Methods that utilize extra side-information to form better, less ambiguous clusters; however, such side-information needs to be provided beforehand and cannot be learned; 2) Methods that can learn from the provided labels to lessen the ambiguity in the formed clusters, but these methods rely on the *cluster assumption* (detailed below), and usually assume that the provided labels are discrete and the *ground truth labels*. This excludes the possibility of learning from indirectly related, but informative side-information. We propose a unified framework that allows using informative side-information directly or indirectly to arrive at better formed clusters. Latent space sharing among different tasks has been studied in a VAE setting (Le et al., 2018; Xie & Ma, 2019). In this work we utilize this latent space sharing framework, but instead focus on clustering with the aid of general, indirect side-information.

**Side-information as constraints**   Using side-information to form better clusters is well-studied. Wagstaff & Cardie (2000) consider both must-link and cannot-link constraints in the context of K-means clustering. Motivated by image segmentation, Orbanz & Buhmann (2007) proposed a probabilistic model that can incorporate must-link constraints. Khashabi et al. (2015) proposed a nonparametric Bayesian hierarchical model to incorporate noisy side-information as soft-constraints. Vu et al. (2019) utilize constraints and cluster labels as side information. Mazumdar & Saha (2017) give complexity bounds when provided with an oracle that can be queried for side information. Wasid & Ali (2019) incorporate side information through the use of fuzzy sets. In supervised clustering, the side-information is the a priori known complete clustering for the training set, which is being used as a constraint to learn a mapping between the data and the given clustering (Finley & Joachims, 2005). In contrast, we do not assume that the constraints are given a priori. Instead, we let the side-information guide the clustering procedure during the training process.

**Semi-supervised methods & the *cluster assumption***   Semi-supervised clustering approaches generally assume that they only have access to a fraction of the true cluster labels. Via constraints as the ones discussed, the available labels are propagated to unlabeled data, which can help mitigate the ambiguity in choosing among different clustering strategies (Bair, 2013). The generative approach to semi-supervised learning introduced in Kingma et al. (2014) is based on a hierarchical generative model with two variational layers. Although it was originally meant for semi-supervised classification tasks, it can also be used for clustering. However, if used for clustering, it has to strictly rely on the *cluster assumption*, which states that there exists a direct correspondence between labels/classes and clusters (Färber et al., 2010; Chapelle et al., 2006). We show that this approach is a special case of our framework without the probabilistic ensemble component (see Sec. 4.2) and when certain distributional assumptions are made. Sansone et al. (2016) proposed a method for joint classification and clustering to address the stringent cluster assumption most approaches make by modeling the cluster indices and the class labels separately, underscoring the possibility that each cluster may consist of multiple class labels. Deploying a mixture of factor analysers as the underlying probabilistic framework, they also used a variational approximation to maximize the joint log-likelihood.

In this work, we generalize the notion of learning from discrete, ground truth labels to learning from indirect, but informative side-information. We make virtually no assumptions on the form of $\mathbf{y}$ nor its relations to the clusters. This makes our approach more applicable to general settings.

## 3 BACKGROUND & PROBLEM SETUP

### 3.1 BACKGROUND—VARIATIONAL DEEP EMBEDDING

The starting point for `DGC` is the *variational auto-encoder* (`VAE`) (Kingma & Welling, 2014) with the prior distribution of the latent code chosen as a Gaussian mixture distribution. This is introduced in Jiang et al. (2017) as `VaDE`. We briefly review the generative `VaDE` approach here to provide the background for `DGC`. We adopt the notation that lower case letters denote samples from their

corresponding distributions; bold, lower case letters denote random variables/vectors; and bold upper case letters denote random matrices.

Assume the prior distribution of the latent code, $\mathbf{z}$, belongs to the family of Gaussian mixture distributions, i.e. $p(\mathbf{z}) = \sum_c p(\mathbf{z}|c)p(c) = \sum_c \pi_c \mathcal{N}(\mu_c, \sigma_c^2 \mathbf{I})$ where $c$ is a random variable, with prior probability $\pi_c$, indexing the normal component with mean $\mu_c$ and variance $\sigma_c^2$. VaDE allows for the clustering of the input data in the latent space, with each component of the Gaussian mixture prior representing an underlying cluster. A VAE-based model can be efficiently described in terms of its generative process and inference procedure. Given an input $\mathbf{x} \in R^d$, the following decomposition of the joint probability $p(\mathbf{x}, \mathbf{z}, c)$ details VaDE's generative process: $p(\mathbf{x}, \mathbf{z}, c) = p(\mathbf{x}|\mathbf{z})p(\mathbf{z}|c)p(c)$. In words, we first sample the component index $c$ from a prior categorical distribution $p(c)$, then sample the latent code $\mathbf{z}$ from the component $p(\mathbf{z}|c)$, and lastly reconstruct the input $\mathbf{x}$ through the reconstruction network $p(\mathbf{x}|\mathbf{z})$. To perform inference and learn from the data, VaDE is constructed to maximize the log-likelihood of the input data $\mathbf{x}$ by maximizing its *evidence lower bound* (ELBO):

$$\log p(\mathbf{x}) \geq \mathbb{E}_{q(\mathbf{z}|\mathbf{x})} \log p(\mathbf{x}|\mathbf{z}) - \mathbb{E}_{q(c|\mathbf{x})} \log \frac{q(c|\mathbf{x})}{p(c)} - \mathbb{E}_{q(\mathbf{z},c|\mathbf{x})} \log \frac{q(\mathbf{z}|\mathbf{x})}{p(\mathbf{z}|c)} \tag{1}$$

where, given the input $\mathbf{x}$, $q(\mathbf{z}, c|\mathbf{x})$ denotes the variational posterior distribution over the latent variables, and $\mathbb{E}_d$ denotes the expectation wrt. *distribution* $d$. With proper assumptions on the prior and variational posterior distributions, the ELBO in Eq. 1 admits a closed-form expression in terms of the parameters of those distributions. We refer readers to Jiang et al. (2017) for additional details.

## 3.2 PROBLEM SETUP

Unlike the unsupervised settings, we *do* assume we have a response variable $\mathbf{y}$, and our goal is to leverage $\mathbf{y}$ to inform a better clustering strategy. Abstractly, given the input-output random variable pair $(\mathbf{x}, \mathbf{y})$, we seek to divide the probability space of $\mathbf{x}$ into non-overlapping subspaces that are meaningful in explaining the output $\mathbf{y}$. In other words, we want to use the prediction task of mapping data points, $x$, sampled from the probability space of $\mathbf{x}$ to their corresponding sampled outcomes $y$ as a *teaching agent*, to guide the process of dividing the probability space of $\mathbf{x}$ into subspaces that optimally explain $y$. Since our goal is to discover the subspace-structure without knowing a priori whether such a structure indeed exists, a probabilistic framework is more appropriate due to its ability to incorporate and reason with uncertainty. To this end, we use and extend the VaDE framework, with the following assumption imposed on the latent code that specifically caters to our setting. Assume the input $\mathbf{x}$ carries predictive information with respect to the

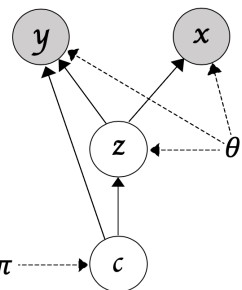

Figure 1: The Bayesian network that underlies the generative process of DGC. $\theta$ and $\pi$ together constitute the generative parameters.

output $\mathbf{y}$. Since the latent code $\mathbf{z}$ should inherit sufficient information from which the input $\mathbf{x}$ can be reconstructed, it is reasonable to assume that $\mathbf{z}$ also inherits that predictive information. This assumption implies that $\mathbf{x}$ and $\mathbf{y}$ are conditionally independent given $\mathbf{z}$, i.e. $p(\mathbf{x}, \mathbf{y}|\mathbf{z}) = p(\mathbf{x}|\mathbf{z})p(\mathbf{y}|\mathbf{z})$.

## 4 DEEP GOAL-ORIENTED CLUSTERING

### 4.1 GENERATIVE PROCESS

In order to incorporate $\mathbf{y}$ into a probabilistic model, recall from our previous discussion that $\mathbf{y}$ might manifest with respect to the input differently across different subspaces of the input space. Viewing $p(\mathbf{y}|\mathbf{z})$ as a conditional probability distribution over $\mathbf{y}$ resulting from a functional transformation from $\mathbf{z}$ to the space of probability distributions over $\mathbf{y}$, we can assume that the ground truth transformation function, $g_c$, is different for each subspace indexed by $c$. If $\mathbf{z} \sim p(\mathbf{z}|c)$ for some index $c$, we assume $p(\mathbf{y}|\mathbf{z}, c) \propto g_c(\mathbf{z})$ for some subspace-specific $g_c$. As a result, we learn a different mapping function for each subspace.

The overall generative process of our model is as follows: **1.** Generate $c \sim \text{Cat}(\pi)$; **2.** Generate $z \sim p(\mathbf{z}|c)$; **3.** Generate $x \sim p(\mathbf{x}|\mathbf{z})$; **4.** Generate $y \sim p(\mathbf{y}|\mathbf{z}, c)$. The Bayesian network that

underlies `DGC` is shown in Fig. 1, and the joint distribution of $\mathbf{x}, \mathbf{y}, \mathbf{z}$, and $c$ can be decomposed as: $p(\mathbf{x}, \mathbf{y}, \mathbf{z}, c) = p(\mathbf{y}|\mathbf{z}, c)p(\mathbf{x}|\mathbf{z})p(\mathbf{z}|c)p(c)$.

## 4.2 Inference & Variational Lower Bound

We first note that the joint variational posterior distribution $q(\mathbf{z}, c|\mathbf{x}, \mathbf{y})$ can be factorized as $q(\mathbf{z}, c|\mathbf{x}, \mathbf{y}) = q(c|\mathbf{x}, \mathbf{z}, \mathbf{y}) \cdot q(\mathbf{z}|\mathbf{x}, \mathbf{y})$. Since we assume we do not have access to the side-information $\mathbf{y}$ at test time, we do not use $\mathbf{y}$ to compute $q(\mathbf{z}|\mathbf{x}, \mathbf{y})$ (in reality, this entails that the encoder only takes $\mathbf{x}$ as input to compute the latent code $\mathbf{z}$). We omit the variable $\mathbf{y}$ in $q(\mathbf{z}|\mathbf{x}, \mathbf{y})$ for the rest of the paper for notation convenience. See Sec. 4.4 for how to compute $q(c|\mathbf{x}, \mathbf{z}, \mathbf{y})$ at test time when $\mathbf{y}$ is not available. With this setup, we have the following variational lower bound (see the Appendix for a detailed derivation)

$$\log p(\mathbf{x}, \mathbf{y}) \geq \underbrace{\mathbb{E}_{q(\mathbf{z}, c|\mathbf{x}, \mathbf{y})} \log p(\mathbf{y}|\mathbf{z}, c)}_{\text{Probabilistic Ensemble}} + \underbrace{\mathbb{E}_{q(\mathbf{z}, c|\mathbf{x}, \mathbf{y})} \log \frac{p(\mathbf{x}, \mathbf{z}, c)}{q(\mathbf{z}, c|\mathbf{x}, \mathbf{y})}}_{\text{ELBO for VAE with GMM prior}} = \mathcal{L}_{\text{ELBO}} \, . \tag{2}$$

The first term in $\mathcal{L}_{\text{ELBO}}$ allows for a probabilistic ensemble of classifiers based on the subspace index. This can be seen as follows

$$\mathbb{E}_{q(\mathbf{z}, c|\mathbf{x}, \mathbf{y})} \log p(\mathbf{y}|\mathbf{z}, c) = \mathbb{E}_{q(\mathbf{z}|\mathbf{x})} \left[ \sum_k \lambda_k \log p(\mathbf{y}|\mathbf{z}, c = k) \right] \approx \frac{1}{M} \sum_{l=1}^{M} \left[ \sum_k \lambda_k \log p(\mathbf{y}|\mathbf{z}^{(l)}, c = k) \right]$$

where $\lambda_k = q(c = k|\mathbf{x})$ and $l$ indexes the Monte Carlo samples used to approximate the expectation with respect to $q(\mathbf{z}|\mathbf{x})$. The probabilistic ensemble allows the model to maintain necessary uncertainty with respect to the discovered subspace structure until an unambiguous structure is captured.

It is also worth noting that the variational lower bound described in Eq. 2 holds regardless of the prior distribution we choose for the latent code $\mathbf{z}$. Although we choose the mixture distribution as the prior in this work, choosing $\mathbf{z} \sim \mathcal{N}(0, \mathbf{I})$ and disregarding the probabilistic ensemble component would recover the exact model introduced in Kingma et al. (2014) (when all labels are missing), and hence is a special case of our proposed framework.

## 4.3 Mean-field Variational Posterior Distributions

Following `VAE` (Kingma et al., 2014), we choose $q(\mathbf{z}|\mathbf{x})$ to be $\mathcal{N}\left(\mathbf{z}|\tilde{\boldsymbol{\mu}}_{\mathbf{z}}, \tilde{\boldsymbol{\sigma}}_{\mathbf{z}}^2 \mathbf{I}\right)$ where $\left[\tilde{\boldsymbol{\mu}}_{\mathbf{z}}, \tilde{\boldsymbol{\sigma}}_{\mathbf{z}}^2\right] = h(\mathbf{x}; \theta)$. $h$ is parameterized by a feed-forward neural network with weights $\theta$. See the Appendix for a detailed discussion of why using a unimodal distribution (i.e. $q(\mathbf{z}|\mathbf{x})$) to approximate a multimodal distribution ($p(\mathbf{z})$) is appropriate in our setting.

Choosing $q(c|\mathbf{x}, \mathbf{z}, \mathbf{y})$ appropriately requires us to analyze the proposed $\mathcal{L}_{\text{ELBO}}$ in greater detail based on the following decomposition (see the Appendix for a detailed derivation):

$$\mathcal{L}_{\text{ELBO}} = \underbrace{\mathbb{E}_{q(\mathbf{z}, c|\mathbf{x}, \mathbf{y})} \log p(\mathbf{y}|\mathbf{z}, c)}_{\text{①}} + \underbrace{\mathbb{E}_{q(\mathbf{z}|\mathbf{x})} \log \frac{p(\mathbf{x}, \mathbf{z})}{q(\mathbf{z}|\mathbf{x})}}_{\text{②}} - \underbrace{\mathbb{E}_{q(\mathbf{z}|\mathbf{x})} \mathbb{KL}\left(q(c|\mathbf{x}, \mathbf{z}, \mathbf{y}) || p(c|\mathbf{z})\right)}_{\text{③}} \, . \tag{3}$$

We observe that since ② does not depend on $c$, $q(c|\mathbf{x}, \mathbf{z}, \mathbf{y})$ should be chosen to maximize $\left(① - ③\right)$. Moreover, the expectation over $q(\mathbf{z}|\mathbf{x})$ does not depend on $c$, and thus has no influence over our choice of $q(c|\mathbf{x}, \mathbf{z}, \mathbf{y})$. Casting finding $q(c|\mathbf{x}, \mathbf{z}, \mathbf{y})$ as an optimization problem, we have

$$\min_{q(c|\mathbf{x}, \mathbf{z}, \mathbf{y})} \quad f_0(q) = \mathbb{KL}\left(q(c|\mathbf{x}, \mathbf{z}, \mathbf{y}) || p(c|\mathbf{z})\right) - \mathbb{E}_{q(c|\mathbf{x}, \mathbf{z}, \mathbf{y})} \log p(\mathbf{y}|\mathbf{z}, c) \, ,$$
$$\text{s.t.} \quad \sum_k q(c|\mathbf{x}, \mathbf{z}, \mathbf{y}) = 1, \quad q(c|\mathbf{x}, \mathbf{z}, \mathbf{y}) \geq 0, \ \forall k \, . \tag{4}$$

The objective functional $f_0$ is convex over the probability space of $q$, as the *Kullback–Leibler divergence* is convex in $q$ and the expectation is linear in $q$. Analytically solving the convex program (4) (see the Appendix for a detailed derivation), we obtain

$$q(c = k|\mathbf{x}, \mathbf{z}, \mathbf{y}) = \frac{p(\mathbf{y}|\mathbf{z}, c = k) \cdot p(c = k|\mathbf{z})}{\sum_k p(\mathbf{y}|\mathbf{z}, c = k) \cdot p(c = k|\mathbf{z})} \, . \tag{5}$$

First we note that since the solution $q(c|\mathbf{x}, \mathbf{z}, \mathbf{y})$ does not depend on $\mathbf{x}$, we omit $\mathbf{x}$ in $q(c|\mathbf{x}, \mathbf{z}, \mathbf{y})$ for the remainder of the paper for notational convenience. To better facilitate understanding, we interpret Eq. 5 in two extremes. If $\mathbf{y}$ is evenly distributed across the different subspaces, i.e. the ground truth transformations $g_c$ are the same for all $c$, then $q(c|\mathbf{z}, \mathbf{y}) = p(c = k|\mathbf{z})$, which is what one would choose for unsupervised clustering (Jiang et al., 2017). However, if the supervised task is informative while the unsupervised task is not, i.e. $p(c|\mathbf{z})$ is a uniform distribution, the likelihoods $\{p(\mathbf{y}|\mathbf{z}, c = k)\}_k$ will dominate $q$. Therefore, one could interpret any in-between scenario as a balance that automatically weights the supervised and the unsupervised tasks based on how strong their signals are with respect to grouping the latent probability space into different subspaces.

### 4.4 EVALUATING ON UNLABELED DATA

We first introduce some notations that we will adopt in this section. We write $p(\mathbf{y}|\mathbf{z}, c)$ to denote the likelihood value, which requires a specific value of $\mathbf{y}$ to compute. We write $p_{\mathbf{y}|\mathbf{z},c}$ to refer to the entire distribution (in the context of the entropy ($\mathbb{H}$) and the expectation ($\mathbb{E}$) operators we refer to distributions, otherwise we refer to specific likelihoods). When presented with the response variable $\mathbf{y}$, Eq. 5 gives the optimal choice of $q(c|\mathbf{z}, \mathbf{y})$ that allows the network to incorporate both the supervised and the unsupervised signals when weighting the clusters. In practice, we do not have access to $\mathbf{y}$ on newly collected (test) data points, which prohibits us from evaluating $q(c|\mathbf{z}, \mathbf{y})$. One easy remedy to this would be to use $p(c|\mathbf{z})$ when $\mathbf{y}$ is not available; however, having an ensemble of well-trained conditional likelihood mappings, $\{p(\mathbf{y}|\mathbf{z}, c = k)\}_k$, and not utilizing them when evaluating on new data points seems wasteful. We thus add a regularization term to $\mathcal{L}_{\text{ELBO}}$, so that DGC can naturally generalize to unlabeled testing samples. The regularized ELBO is:

$$\mathcal{L}_{\text{ELBO}}^{\textbf{regu}} = \mathcal{L}_{\text{ELBO}} - \mathbb{E}_{q(\mathbf{z},c|\mathbf{x},\mathbf{y})}\mathbb{H}_{\textbf{max}}(p_{\mathbf{y}|\mathbf{z},c}) \,, \tag{6}$$

where $\mathbb{H}_{\textbf{max}}(p_{\mathbf{y}|\mathbf{z},c}) = \textbf{max}\{\mathbb{H}(p_{\mathbf{y}|\mathbf{z},c}), 0\}$ and $\mathbb{H}(p_{\mathbf{y}|\mathbf{z},c}) = -\mathbb{E}_{p_{\mathbf{y}|\mathbf{z},c}} \log p_{\mathbf{y}|\mathbf{z},c}$, which is the entropy of the task network distributions. If $\mathbf{y}$ is a discrete random variable, $\mathbb{H}_{\textbf{max}}(p_{\mathbf{y}|\mathbf{z},c}) = \mathbb{H}(p_{\mathbf{y}|\mathbf{z},c})$, which is the entropy of $p_{\mathbf{y}|\mathbf{z},c}$ and always non-negative; on the other hand, when $\mathbf{y}$ is continuous, although the differential entropy of $p_{\mathbf{y}|\mathbf{z},c}$ can take any sign, the $\textbf{max}$ operator ensures that $\mathbb{H}_{\textbf{max}}(p_{\mathbf{y}|\mathbf{z},c})$ will remain non-negative. Therefore, adding (a convex combination of) negative entropies preserves the inequality, and thus $\mathcal{L}_{\text{ELBO}}^{\textbf{regu}}$ remains a proper lower bound. Additionally, solving a similar convex program provides the optimal choice of $q(c|\mathbf{z}, \mathbf{y})$ in the presence of the regularizer

$$q(c = k|\mathbf{z}, \mathbf{y}) = \frac{e^{\log p(\mathbf{y}|\mathbf{z},k) - \mathbb{H}_{\textbf{max}}(p_{\mathbf{y}|\mathbf{z},k})} \cdot p(k|\mathbf{z})}{\sum_j e^{\log p(\mathbf{y}|\mathbf{z},j) - \mathbb{H}_{\textbf{max}}(p_{\mathbf{y}|\mathbf{z},j})} \cdot p(j|\mathbf{z})} \,. \tag{7}$$

This form of regularization penalizes the entropies of the conditional distributions $p_{\mathbf{y}|\mathbf{z},c}$. More specifically, clusters with higher posterior weights are penalized more towards having low entropies. This aligns with our intuition: the most suitable cluster to explain a given sample $y$ should be relatively more certain in how it is distributed. We thus use Eq. 7 to weight the clusters during training. It is worth noting that although Eq. 7 provides the optimal choice of $q(c|\mathbf{z}, \mathbf{y})$ for maximizing $\mathcal{L}_{\text{ELBO}}^{\textbf{regu}}$, any choice of $q(c = k|\mathbf{z}, \mathbf{y})$, as long as it maintains a proper probability distribution, would satisfy the fact that $\mathcal{L}_{\text{ELBO}}^{\textbf{regu}}$ is a proper lower bound. Based on the previously stated intuition, when evaluating on an unlabeled data point, we use

$$q^{\text{test}}(c = k|\mathbf{z}, \mathbf{y}) = \frac{e^{-\mathbb{H}_{\textbf{max}}(p_{\mathbf{y}|\mathbf{z},k})} \cdot p(k|\mathbf{z})}{\sum_j e^{-\mathbb{H}_{\textbf{max}}(p_{\mathbf{y}|\mathbf{z},j})} \cdot p(j|\mathbf{z})} \tag{8}$$

to weight the clusters. This aligns with our previous reasoning: the cluster that corresponds to $p(\mathbf{y}|\mathbf{z}, c)$ with the lowest entropy will be weighted most heavily. This allows the model to use the tuned conditional likelihood mappings when the side-information $\mathbf{y}$ is not available.

## 5 EXPERIMENTS

We investigate the efficacy of DGC on a range of datasets. We refer the reader to the Appendix for the experimental details, e.g. the train/validation/test split, the chosen network architecture, the choices of learning rate and optimizer.

### 5.1 NOISY MNIST

We introduce a synthetic data experiment using the MNIST dataset, which we name the *noisy MNIST*, to illustrate that the supervised part of DGC can enhance the performance of an otherwise well-performing unsupervised counterpart. Further, we explore the behavior of DGC without its unsupervised part to demonstrate the importance of capturing the inherent data structure. We extract images that correspond to the digits 2 and 7 from MNIST. For each digit, we randomly select half of the images for that digit and superpose noisy backgrounds onto those images (see the Appendix for image samples). The binary random variable **y** indicates what digit each image belongs to. Our goal is to cluster the images into 4 clusters: digits 2 and 7, with and without background. However, we are only using the binary responses for supervision and have no direct knowledge of the background. We therefore parameterize the task networks, $\{p(\mathbf{y}|\mathbf{z}, c = k)\}_{k=1}^4$, as Bernoulli distributions where we learn the parameters (the probabilities).

As a baseline, the unsupervised approach, VaDE, already performs well on this dataset, achieving a clustering accuracy of 95.6% when the desired number of clusters is set to 4. Fig. 2a shows that VaDE distinguishes well based on the presence or absence of the noisy background, and incorrectly clustered samples are mainly due to VaDE's inability to differentiate the underlying digits. This is reasonable behavior: if the background signal dominates, the network may focus on the background for clustering as it has no explicit knowledge about the digits.

DGC performs nearly perfectly (with a clustering accuracy of 99.6%) in this setting with the help of the added supervision. We see that DGC mitigates the difficulty of distinguishing between digits under the presence of strong, noisy backgrounds (as shown in Fig. 2b, where DGC makes almost no mistakes in distinguishing between digits even in the presence of noisy backgrounds). This added supervision does not overshadow the original advantage of VaDE (i.e. distinguishing whether the images contain background or not). Instead, it enhances the overall model in cases where the

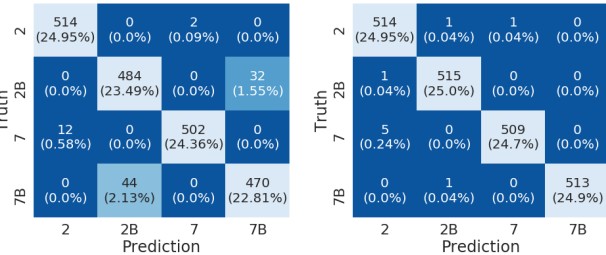

(a) Confusion Matrix—VaDE     (b) Confusion Matrix—DGC

Figure 2: Confusion matrices abbreviated, 2B/7B, in the row/column labels denotes digits 2/7 with background. Rows represent the predicted clusters, and columns represent the ground truth.

unsupervised part, i.e. VaDE, struggles. Furthermore, as detailed in Sansone et al. (2016) and earlier sections, most existing approaches that take advantage of available labels rely on *the cluster assumption*, which assumes a one-to-one correspondence between the clusters and the labels used for supervision. This experiment is a concrete example that demonstrates DGC does not need to rely on such an assumption to form a sound clustering strategy. Instead, DGC is able to work with class labels that are only partially indicative of what the final clustering strategy should be, potentially making DGC more applicable to more general settings.

**Ablation study** To further test the importance of each part of our model, we ablate the probabilistic components (i.e. we get rid of the decoder and the loss terms associated with it, so that only the supervision will inform how the clusters are formed in the latent space) and perform clustering using only the supervised part of our model. We find that clustering accuracy degrades from the nearly-perfect accuracy obtained by the full model to 50%. Coupled with the improvements over VaDE, this indicates that each component of our model contributes to the final accuracy and that our original intuition that supervision and clustering may reinforce each other is correct.

### 5.2 PACMAN

In this experiment we test DGC's ability to learn a clustering strategy when facing a *continuous* response as *side-information*. The Pacman-shaped data consists of two annuli and each point in the two annuli is associated with a continuous response value (see the Appendix for a detailed breakdown). These response values decrease linearly (from 1 to 0) in one direction for the inner (yellow) annulus, and increase exponentially (from 0 to 1) in the opposite direction for the outer

(purple) annulus. Figure 3a contains a 3D illustration of the dataset. We use linear/exponential rates for the responses to not only test our model's ability to detect different trends, but also to test its ability to fit different rates.

Our goal is to separate the two annuli depicted in Fig. 3a. This is challenging as the annuli were deliberately chosen to be very close to each other. We applied various traditional unsupervised learning methods including K-means and hierarchical clustering to only the 2D Pacman-shaped data (i.e., not using the responses, but only the 2D Cartesian coordinates). Besides hierarchical clustering with single linkage (and not other distance metric), none of the unsupervised methods managed to separate the two annuli. Moreover, these approaches also result in different clustering strategies as they are based on different distance metrics (see the Appendix for these results). This phenomenon echos a deep-rooted obstacle for clustering methods in general: the concept of clustering is inherently subjective, and different distance metrics can potentially produce different, but sometimes equally meaningful, clustering results.

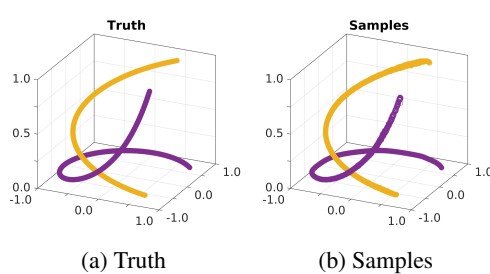

(a) Truth       (b) Samples

Figure 3: (a) The ground truth 3D Pacman; (b) The generated samples from `DGC`.

Applying `DGC` with the input $x$ as the 2D Cartesian point coordinates and the responses $y$ as the response values described previously, we are able to distinguish the two annulli wholly based on the discriminative information carried by the responses. We parameterize the task networks, $\{p(\mathbf{y}|\mathbf{z}, c = k)\}_{k=1}^{2}$, as Gaussian distributions where we learn the means and the covariance matrices. As the generated samples from Fig. 3b shows, both the Pacman shape and its corresponding response values are captured.

Table 1: Test clustering accuracies on the Pacman dataset

| Models | Test Clustering Accuracy |
| --- | --- |
| VaDE | $50.4\% \pm 0\%$ |
| NN-DGC | $81.6\% \pm 5.3\%$ |
| AUG-SS | $82.3\% \pm 4.6\%$ |
| DGC | $\mathbf{93.5\% \pm 3.9\%}$ |

The generated samples from `DGC` substantiate the model's ability to appropriately learn and use the side-information provided by the response values to obtain a sensible clustering strategy. Unlike most previously discussed methods, `DGC` can work with continuous response values. This is highly attractive, as it lends itself to any general regression setting in which one would believe the desired clustering strategy should be informed by the regression task.

Finally, we compare `DGC` to `VaDE`, its ablated version, and a baseline method to substantiate the efficacy of our proposed framework. First, although the solution to the convex programming in Eq. 4 provides an optimal choice of $q(c|\mathbf{z}, \mathbf{y})$ from a theoretical standpoint, our proposed framework, specifically the proposed $\mathcal{L}_{\text{ELBO}}$ (Eq. 2), holds for any choice of $q(c|\mathbf{z}, \mathbf{y})$. We thus ablate the convex programming component of our model and parameterize $q(c|\mathbf{z}, \mathbf{y})$ using a neural network (NN-DGC). Second, by choosing $\mathbf{z} \sim \mathcal{N}(0, \mathbf{I})$, the unsupervised part of `DGC` recovers exactly the semi-supervised (SS) approach introduced by Kingma et al. (2014) in the case when all labels are missing. Since SS is not expected to perform well in a purely unsupervised setting, we include the probabilistic ensemble component as an augmentation (AUG-SS). The results described in Tab. 1 are obtained from running each model 100 times, and demonstrate the following: 1) without the additional responses $\mathbf{y}$, `VaDE` cannot distinguish between the two annuli at all, emphasizing the importance of exploiting the additional information; 2) the convex programming (Eq.4) is crucial to the success of `DGC` and it is difficult for a neural network to find the same optimal distribution; 3) the choice of the prior on the latent code $\mathbf{z}$ is also of paramount importance, and the Gaussian mixture distribution is more suitable for modeling clusters than an isotropic Gaussian.

### 5.3   SVHN

We apply `DGC` to the Street View House Number (SVHN) dataset (Netzer et al., 2011) where the digit labels (10 digits in total) are used as the ground truth clustering labels. This dataset consists of 73,257 training images, 26,032 test images, and 531,131 additional training images. We train `DGC` using all the training and extra images. We parameterize the task networks, $\{p(\mathbf{y}|\mathbf{z}, c = k)\}_{k=1}^{10}$, as multinomial distributions over the 10 digits where we learn the parameters of those distributions.

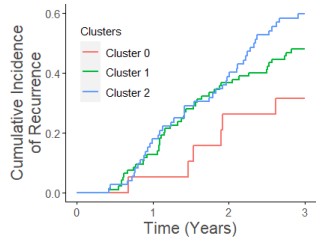

(a) The Kaplan-Meier curves for `DGC`

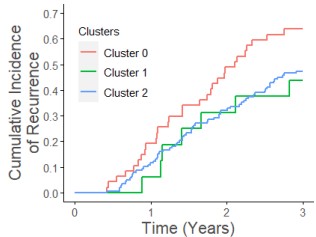

(b) The Kaplan-Meier curves for `VaDE`

Figure 4: The Kaplan-Meier risk differences curves among clusters from `DGC` and `VaDE`.

The goal of this experiment is to investigate the impact of the hyperparameter, the number of clusters desired, on the overall framework. Ideally one would hope that a model would effectively ignore additional clusters when the number of chosen clusters is larger than the number of ground truth clusters. Instead, it should automatically determine how

Table 2: Clustering (and classification) accuracy variation over maximum possible clusters.

| Clusters | DGC | K-means | VaDE | AUG-SS |
|---|---|---|---|---|
| **10** | **92.6** (**94.7**) | 88.0 | 28.7 | 78.8 (80.2) |
| 20 | 91.9 (92.2) | 56.1 | 16.4 | 76.2 (71.3) |
| 50 | 87.9 (91.3) | 27.3 | 14.7 | 74.3 (68.2) |
| 100 | 84.1 (89.3) | 17.0 | 9.3 | 71.3 (64.9) |

many clusters are appropriate during learning, which is usually unknown a priori. To test if `DGC` has this property, we alter the desired number of clusters. We use the class labels as the responses to help the clustering. It is worth noting that while in general one will not use ground-truth labels as side-information, we use them for this experiment as they provide a good idea of what a reasonable number of clusters in latent space should be.

Table 2 demonstrates the clustering and classification accuracies obtained from `DGC` and various baselines over a varying number of underlying clusters. The results indicate that `DGC` most successfully groups the data into the proper classes and clusters regardless of the number of clusters available. Firstly, note that `DGC` outperforms `VaDE` and AUG-SS by significant margins, demonstrating the importance of the added side-information and the Gaussian mixture prior assumption, respectively. Additionally, although we use the ground truth labels as the side-information, `DGC` outperforms the K-means baseline, where we perform the K-means clustering on the features obtained from the last hidden layer of a classification network that is trained on the SVHN dataset (with a 95.7% classification accuracy). This demonstrates the advantages of the latent space sharing and the end-to-end nature of `DGC`. Secondly, even when the desired number of clusters is larger than the number of digits, a network may still achieve a clustering accuracy of 100% if it learns to group samples into a consistent set of clusters, with the cardinality of that set matching the number of digits. This is echoed by the clustering accuracies shown in Tab. 2. When the desired number of clusters exceeds 10, `DGC` is still able to achieve high clustering accuracies. By comparison, the clustering accuracy drops dramatically when the number of desired clusters increases for all baselines, demonstrating `DGC`'s ability to choose an appropriate number of clusters.

## 5.4 CAROLINA BREAST CANCER STUDY (CBCS)

In this experiment we apply `DGC` to a real-world breast cancer dataset collected as part of the Carolina Breast Cancer Study (CBCS). The dataset consists of 1,713 patients, each of which has 2-4 associated histopathological images and a list of biological markers. e.g., the Pam50 gene expressions (Troester et al., 2018) and ER status.

As an exploratory investigation, we use the binary indicator for breast cancer recurrence as the response variable $\mathbf{y}$. Applying deep learning techniques, supervised or unsupervised, to analyze histopathological images of breast cancer has gained traction in recent years (Xie et al., 2019). Distinguished from those methods, our goal is to inspect whether the discovered clusters, whose formation is influenced both by the supervised recurrence information and the unsupervised reconstruction signal, carry meaningful information in terms of survival rate or gene expression. We parameterize

the task networks, $\{p(\mathbf{y}|\mathbf{z}, c = k)\}_{k=1}^{3}$, as Bernoulli distributions and learn the associated parameters. See the Appendix for experimental details.

To investigate whether the three clusters that we discovered were identifying meaningful differences in tumor biology, we examine the differences in rates of cancer recurrence and features of tumor aggressiveness between the clusters. We also compare to the baseline clusters obtained from the purely unsupervised `VaDE` to corroborate the importance of the added side-information.

Using a Kaplan-Meier estimator to estimate risk differences for time to cancer recurrence within three years, we obtained a p-value of 0.0024 and observed that Cluster 0 had the lowest risk of recurrence and Cluster 2 had the highest risk (see Tab. 3). Even with the small sample size, we observed substantial differences in recurrence risk at three years of follow-up between the clusters, particularly

Table 3: The risk of recurrence difference (RRD) between clusters for `DGC`.

| Comparsion | RRD (95% CI) |
|---|---|
| Cluster 0 VS Cluster 1 | 16.3% (-6, 39) |
| Cluster 0 VS Cluster 2 | 30.0% (5, 55) |

Clusters 0 and 2 (see Fig. 4a). By comparison, the differences in recurrence risk between the clusters from `VaDE` is much less significant, both visually (see Fig. 4b, where two clusters almost overlap) and in terms of p-value (0.073).

Comparing tumor characteristics, we observed that Cluster 0 contained more indolent tumors, characterized by good-prognosis features such as estrogen-receptor (ER) positivity, low grade, and Luminal A tumor subtype (see Tab. 4). In contrast, more aggressive tumor characteristics were featured in Clusters 1 and 2, such as negative ER status, high grade, and Basal-like tumor subtype, although Cluster 1 appeared to be intermediate between Cluster 0 and 2 in some characteristics. Coupled with the differences in cancer outcomes, these differences in tumor characteristics indicate that the

Table 4: Tumor characteristics for each cluster. Features are color-coded as low , intermediate , or high risk.

| | | Cluster 0 N(%) | Cluster 1 N(%) | Cluster 2 N(%) |
|---|---|---|---|---|
| ER Status | Positive | 15 (78.9) | 53 (56.4) | 42 (57.5) |
| | Negative | 4 (21.1) | 41 (43.6) | 31 (42.5) |
| Grade | Low | 7 (36.8) | 14 (14.9) | 4 (5.5) |
| | Medium | 5 (26.3) | 25 (26.6) | 25 (32.4) |
| | High | 7 (36.8) | 55 (58.5) | 44 (60.3) |
| Tumor Subtype | Luminal A | 10 (55.6) | 27 (29.0) | 16 (21.9) |
| | Luminal B | 6 (33.3) | 17 (18.3) | 19 (26.0) |
| | ER-/HER2+ | 1 (5.5) | 8 (8.6) | 3 (4.1) |
| | Basal-like | 1 (5.5) | 41 (44.1) | 35 (47.9) |

method successfully distinguished between tumors with low-risk features (Cluster 0) and tumors with intermediate- and high-risk features (Clusters 1 and 2). We include the same table that characterizes tumor characteristic for clusters obtained from `VaDE` in the Appendix. As one can see, cluster 0 from `VaDE`, which has the highest recurrence rate, should have the most negative ER subtype, the most high grade, and the most Basal-like tumor subtype. As for grade, it is not the cluster with the most high grade patients. For ER status and tumor subtype, it does have the highest negative ER subtype and the most Basal-like tumor subtype, but the differences are much less significant compared to the clusters obtained from `DGC`. This indicates that using recurrence side-information, via our `DGC` approach, indeed resulted in more meaningful clusters.

## 6 CONCLUSION

In this work, we introduced `DGC`, a probabilistic framework that allows for the integration of both supervised and unsupervised information when searching for a congruous clustering in the latent space. This is an extremely relevant, but daunting task, where previous attempts are either largely restricted to discrete, supervised, ground-truth labels or rely heavily on the side-information being provided as manually tuned constraints. To the best of our knowledge, this is the first attempt to simultaneously learn from generally indirect, but informative side-information and form a sensible clustering strategy, all the while making minimal assumptions on either the form of the supervision or the relationship between the supervision and the clusters. This method is applicable to a variety of fields where an instance's input and task are defined but its membership is important and unknown, e.g., survival analysis. Training the model in an end-to-end fashion, we demonstrate that `DGC` is capable of capturing a clustering that aligns with the provided information, while obtaining reasonable classification results on various datasets.

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
