# OpenReview forum: "Deep Goal-Oriented Clustering"
_ICLR.cc/2021/Conference — Reject_

### Official Review · AnonReviewer1 · 2020-10-15
**Not a realistic modeling**

**Rating:** 3
**Confidence:** 4

**Review:**

I find the problem setup a bit superficial and I was not convinced from the examples in the experiment section  that this is a meaningful real world situation.

What is the parametric modeling of  p(y|z,c)?

What is the parametric modeling of q(y|x)? Is it actually q(y|\mu_z) where mu_z = E(z|x)?

In (6) do you have a tunable regularization coefficient for the regularization term?

The notation in (5) is odd. z appears on the right side of the equation but not on the left side.

In the last experiment (medical data) you show clustering results  but their is no reference in the results to the recurred binary indicator. We dont know whether  this information improved the clustering.

---

> ### Author Response · Authors · 2020-11-18
> **Thank you for your comments!**
>
> Thank you for your valuable comments and questions. We answer your questions in the same order as they were posed
>
> ### Q1: What is the parametric modeling of p(y|z,c)?
> 1. We apologize as we should have mentioned in the paper that the parametric modeling of p(y|z,c) is chosen to be Gaussian, where the mean and the covariance matrix are learned, when y is continuous, and is chosen to be multinomial when y is discrete. We have updated this information to all the experiments in the updated manuscript.
>
> ### Q2: What is the parametric modeling of q(y|x)?
> 2. Could you clarify what distribution you are referring to? We do not have q(y|x) in this work; however, we do have two variational posteriors q(z|x,y) and q(c|z,x,y) (see Common Concern #1 for the notation change). The former is chosen to be Gaussian where the mean and the covariance matrix are learned, and the later is the solution to the convex program ( (4) in Sec.4.3).
>
> ### Q3: Is there a tunable regularization coefficient in Eq. (6)?
> 3. Thank you for this question. We do not have a tunable regularization coefficient in (6) in the current version of our work, although one can be easily added while preserving the fact that (6) is still a proper lower bound of the joint log-likelihood, log p(x,y). We will certainly explore this further in future iterations of this work.
>
> ### Q4: Notation in (5) is inconsistent
> 4. We apologize about the confusion regarding the notations in those equations. We have fixed this inconsistency by making a different assumption on the joint variational posterior distribution. Please refer to the overall comment about (5), (7), and (8) (see the Common Concern #1 in the overall comment).
>
> ### Q5: There is no comparison in the CBCS experiment that demonstrates the side-information improves clustering
> 5. Thank you for this great point! We certainly agree that the efficacy of our proposed model could be much better assessed if we include a baseline comparison for the CBCS dataset. We have thus included the results obtained from just applying the unsupervised VaDE to the CBCS dataset. Please refer to the overall comment about the added comparison and analysis for the CBCS dataset (see Common Concern #2 in the overall comment).

---

### Official Review · AnonReviewer2 · 2020-10-23
**A clustering method that uses an additional response variable as side information**

**Rating:** 4
**Confidence:** 4

**Review:**

The paper proposes a method called DGC (Deep Goal-Oriented Clustering) for clustering using side information in the form of a response veriable y. Therefore the objective is to cluster data objects x given a response variable y for each x, that may not be explicitely related to the cluster structure.

The approach is based on probabilistic modeling and constitutes an extension of VaDe with the addition of the response variable y. It performs end-to-end clustering while simultaneously learning from the available side information.

There are major issues to be addressed.
1) It is a strict requirement that a response y should be available for each data point x. Typically in constrained clustering only a small set of constraints is availabe.
2) If y is available for each x, then we could perform typical clustering using the augmented vectors (x,y). This is a trivial baseline for comparison.
3) In experiments with the SVHN dataset, ground truth labels are used as side information. In such a case the problem can be easily solved by training a neural classifier that for each x predicts the class y. If we wish to cluster x, the last hidden layer embeddings could be used for clustering.
4) In eq. (5) and eq. (8) the formulas for computing q(c=k|x) do not involve x, but rather involve z. This inconcistency should be fixed or clarified.
5) The discussion beyond eq. (5) is not clear.
6) It is not clear how to compute eq. (8) since y is not available.
7) A pseudocode of the proposed algorithm should be provided in order for several details to be clarified.

---

> ### Author Response · Authors · 2020-11-18
> **Thank you for your comments!**
>
> Thank you for your valuable comments and concerns. We address your concerns in the order as they were posed
>
> ### Q1: It is a constraint that y should be available for each data point
>
> 1. In typical constrained clustering, the constraints usually provide DIRECT information about how clusters are formed. By comparison, the side-information y in our case is assumed to carry indirect, but useful information about how clusters should be formed. Such indirect information typically will be available for each data point x, and our method is designed to incorporate such information when it is available. We additionally want to emphasize the following two points
>
> 	(1). We only use the side-information y during training, and DO NOT make use of it during test time (see Sec. 4.4).
>
> 	(2). In fact, our method can even handle the scenario in which the side-information is missing for some observations x during training. One can simply treat observations with missing side-information y during training as unlabeled data, and use the entropy to calculate q(c|z,x,y) (Common Concern #1 for notation change & Sec. 4.4). Notwithstanding, we decided to explore this aspect of DGC (that it can be applied in a semi-supervised setting) for future extension of this work, and instead focus on the case where the side-information y is not missing during training.
>
> ### Q2: Perform typical clustering using the augmented vectors (x,y)
> 2. This is indeed a great suggestion for baseline for the case where we have access to y during training and testing. We did not use this as a baseline because we assume we do not have access to y during test time. For example, in our breast cancer experiment we use recurrence information as the side-information during training. However, for a new patient (i.e. at test time) we want to know if he/she falls into a subgroup with different survival characteristics without having the recurrence information apriori. (This could, for example, inform different types of treatment choices.) In other words, for this patient we do not know if the cancer will recur or not. This scenario was what motivated the development of DGC in the first place. We now state this more clearly in the manuscript.
>
> ### Q3: Perform clustering on the features extracted from the last hidden layer of a classification network
> 3. Thank you for suggesting this baseline comparison. We now include this baseline comparison in the updated paper. We now provide the clustering result from applying the k-means algorithm on the features obtained from the last hidden layer of a classification network (that achieved a classification accuracy of 95.8%). As you can see in Table 2, DGC outperforms this baseline for every choice of the number of clusters desired.  Please also refer to the Common Concern #3 in the overall comment for an explanation of our initial motivation for conducting this experiment on the SVHN dataset.
>
> ### Q4: Notations in (5) and (8) are inconsistent
> 4. We have addressed this inconsistency in the updated paper. Please refer to the Common Concern #1 in the overall comment for a high-level description of our proposed fix.
>
> ### Q5: Writing is not clear beyond Eq. (5)
> 5. We have attempted to explain things in a more explicit manner beyond Eq. (5) in the updated version. Please let us know if there is still anything that remains unclear.
>
> ### Q6: It is not clear how to compute Eq. (8) when y is not available
> 6. We apologize for the confusion that Eq. (8) causes. However, please note that Eq. (8) ONLY requires the ENTROPY of the task networks, \{ p(y|z,c=k \}_{k=1}^K, where K is the number of clusters desired (a hyperparameter set by the user) to compute. In other words, its computation does not require specific values of y. We have attempted to clarify the notation further in the updated version, so please let us know if you still find it confusing or unclear.
>
> ### Q7: A pseudocode of the proposed algorithm should be added
> 7. Thank you for this suggestion. We will add a pseudocode of the proposed algorithm in the updated version of the paper.

---

### Official Review · AnonReviewer3 · 2020-10-27
**Clearly explained, intuitive approach for clustering+prediction, concerns over technical and experimental depth.**

**Rating:** 5
**Confidence:** 3

**Review:**

**Summary:**

This paper introduces Deep Goal Oriented Clustering, an approach for joint clustering and classification. The approach shares a latent embedding for the data between the two tasks. The latent embedding is parameterized by a mixture of Gaussians. The approach gives a probabilistic, VAE-based formulation and derives the variational lower bound for the model. The authors run experiments investigating the effectiveness of their approach and the impact of the clustering-component of their approach.

Summary of Review: Clearly explained, intuitive approach for clustering+prediction, concerns over technical and experimental depth.

**Strengths:**

The presentation of the paper is quite clear. The authors take care to explain their model intuitively and provide with the readers guidance In both high level formulation as well as details of the technical approach and the experiments performed. The model presented is simple and intuitive. The claims of the paper are generally well supported with the explained experiments. The authors provide experiments both on synthetic data as well as data from an ongoing breast cancer study.

**Weaknesses:**

**Technical Novelty**: Sharing latent space representations for clustering and classification in these VAE-based models is not, as I understand, particularly novel. There exist previous works that present quite similar approaches. For instance Le et al, (2018) uses a very similar architecture as do Xie and Ma (2019). The idea of exploring the "cluster" assumption and using a mixture model latent variable is quite interesting and distinct from previous work, but I feel that the current form of this contribution is somewhat lacking in terms of analytical and experimental depth.

**Experimental Results**: I have some additional concerns over the level of competition compared to state-of-the-art in the empirical results of the paper. My understanding is that the approach is not state-of-the-art compared to other methods on the SVHN dataset (as compared to the error rates listed on this leader board [1]). My understanding is that noisy MNIST is rather limited in scope and in its scale (in number of classes). Unless I have missed something, DGC is not compared to other methods on the breast cancer data?

**Comparison to Semi-Supervised VAE (M2)**: The set up of the proposed DGC graphical model is closely related to but distinct from that of the semi-supervised VAE. The authors discuss their relationship with respect to the cluster assumption, but I was eager to see additional insight as to why (or when) one model's dependencies for Y would be advantageous.


[Le et al, 2018] Lei Le, Andrew Patterson and Martha White. Supervised autoencoders: Improving generalization performance with unsupervised regularizers. NeurIPS 2018. https://papers.nips.cc/paper/7296-supervised-autoencoders-improving-generalization-performance-with-unsupervised-regularizers.pdf

[Xie and Ma, 2019] Zhongbin Xie and Shuai Ma. Dual-View Variational Autoencoders for Semi-Supervised Text Matching https://www.ijcai.org/Proceedings/2019/0737.pdf

[1] https://paperswithcode.com/sota/image-classification-on-svhn

**Questions for the authors**

* What is the hierarchical clustering method used on the Pacman data? It seems like something like Single Linkage may  be able to correctly recover this data?

---

> ### Author Response · Authors · 2020-11-18
> **Thank you for your comments!**
>
> Thank you for your valuable comments, suggestions, and concerns. We address your concerns and question in the order as they were posed
>
> ### Technical Novelty:
>  Thank you for pointing out previous works that focus on latent space sharing among different tasks in a VAE setting. We are aware that works on latent space sharing already exist, and did not intend to claim this was one of our contributions. Nevertheless, we added the references you listed in the related work section. As you astutely noted, our contributions lie in the fact that our model is able to utilize general side-information y in a probabilistic manner, and explore the “cluster assumption” in a VAE + Gaussian Mixture Prior setting.
>
> ### Experimental Results:
>  Thank you for your comments. Please refer to the Common Concern #3 in the overall comment for an explanation of our initial motivation for conducting the SVHN experiment. We also want to clarify that we focus on clustering in this experiment instead of classification, as in general one would not use ground truth labels as side-information. Therefore, the purpose of this experiment is not to compete with the state-of-the-art classification results on the SVHN dataset. Last but not least, we have added an additional comparison to the clustering results obtained from applying VaDE on the CBCS dataset. Please refer to the Common Concern #2 in the overall comment.
>
> ### Comparison to Semi-Supervised VAE (M2):
> As you accurately pointed out, DGC is not constrained by the cluster assumption as semi-supervised VAE (M2) would be if used for clustering. Moreover, please refer to point (2) of the Clarification on DGC VS Semi-Supervised VAE (M2) in the overall comment for a description of a deeper connection between DGC and semi-supervised VAE (M2).
>
>
> ### Answer to the Question:
> We used the hierarchical clustering with the Ward distance in the Appendix. After seeing this question, we ran hierarchical clustering with single linkage on the Pacman data and it indeed captured the two clusters. We added this result to the updated version of the Appendix. Nonetheless, we do not think this contradicts the message we are trying to convey, as in reality one will hardly ever have such accurate prior knowledge on what distance metric is the most appropriate. Such choices of simple distance measurements become much more complicated when it comes to high-dimensional data like images or audios.

---

### Official Review · AnonReviewer4 · 2020-10-28
**Nice ideas that would benefit from more experimental validation**

**Rating:** 6
**Confidence:** 4

**Review:**

### Summary

In traditional clustering algorithms, incorporation of “side-information”, or additional features only available during training time, typically assume some prior knowledge of the ground-truth clusters, or constraints on those clusters. However, this need not be the case, as training samples may contain arbitrary information which only indirectly corresponds to true cluster labels. This work proposes a novel method, called Deep Goal-Oriented Clustering (DGC), to incorporate such arbitrary “side-information” into a probabilistic auto-encoder based clustering algorithm.

The basis of this model is a traditional deep clustering algorithm, Variational Deep Embeddings (VaDE) - A deep auto-encoder that models the latent space with a Gaussian Mixture Model prior. The authors augment this fully unsupervised model to be able to additionally generate side information based on the latent gaussian class, and the latent code. This formulation allows the model to incorporate the information of $y$ into the posterior distribution over classes, but does not require any type of correspondence between $y$ and the clusters that will be generated.

The authors provide a description of a mean-field variational approximation to maximize the likelihood of the training samples containing $x$ and $y$. They show an analytical solution to find $q(c | x)$ given x and y, and provide intuitions backing this solution. Finally, they show a novel formulation to take advantage of the conditional likelihood $p(y | z, c)$ and incorporate it into the posterior $q(c | x)$, when $y$ is not available during test-time.

The authors corroborate their methods on 4 separate tasks, in total.
The first is an augmented MNIST dataset, which consists of 4 general classes of images: images of 2 and 7, with and without noisy background. During training, the ground-truth digit is provided as side-information, but no information is provided about the background information. The results demonstrate that DGC can leverage this additional side-information to vastly outperform VaDE, which cannot incorporate such additional information.
The second task is a synthetic PACMAN dataset, consisting of 2 very closely placed 2-D annuli with a 3rd response feature which is generated via either linear or exponential rates. Here, the input is the 2-D coordinates of a point in either annuli, and the side-information is the continuous response feature. The authors demonstrate that, without incorporating the side information, correctly clustering points from either annuli is difficult for any clustering algorithm. However, including the 3rd response feature into DCG allows the model to correctly separate and model both annuli.
The third dataset is the Street View House Number dataset, another digit recognition dataset. The authors use this dataset to explore the effect of changing the number of clusters, despite having a fixed number of ground-truth clusters, for which labels are provided as side-information.
Finally, the authors examine the Carolina Breast Cancer Study, for which the demonstrate that DCG is able to utilize the side-information of whether or not a cancer recurred to separate instances of patients into 3 different, meaningful risk categories.

### Quality

The motivation and methods are explained well and the experiments corroborate the usefulness of their method. However, some of the experiments could benefit from more salient baseline comparisons. For example, both the SVHN and CBCS experiments would benefit if the results from DCG were compared again to VaDE to demonstrate the effect of incorporating side-information. This is especially true for CBCS as the authors argue early in the work that DCG should improve the ambiguity of the learned clusters; without the VaDE baseline, improvement of ambiguity cannot be demonstrated. Additionally, for a dataset like SVHN where the side-information is the ground-truth labels, it would be useful to compare to a semi-supervised baseline, such as Kingma et. al, 2014, to demonstrate how DCG performs compared to models which make stronger assumptions about the side-information when those stronger assumptions are valid.

### Clarity

Overall the paper is clear and well-written. The motivation would benefit from a few real-world examples, to help distinguish the case where side-information fits under the cluster assumption and where it does not, but otherwise the motivation for the method is well-explained. The description and background of the model is precise and easy to understand. The only section I find lacking in terms of clarity is the section on SVHN - there is little description of the dataset itself, what the classes are, how classification is computed, or why the hyper-parameter of the number of clusters is an interesting hyper-parameter to ablate in this task. Also, some experimental details that are in the appendix should be moved to the main text, such as how significance is assessed and how validation was performed, which are important to interpret the results.

### Originality

Overall, the method is a somewhat simple extension of VaDE. However, the extension is a novel exploration of incorporating side-information into a probabilistic model, and the emphasis on making as few assumptions as possible about the side-informations’ relevance to the true cluster labels is interesting. The following works were not cited but perhaps should be:

- Density-based clustering with side-information and active learning, Vu and Do, 2017: incorporate both constraint information and cluster labels jointly as side-information.

- Query Complexity of  Clustering with Side-Information, Mazumda and Sahar, 2017: Prove some theoretical bounds on the complexity of clustering when you can ask an oracle for side-information (as constraints) on a pair of samples at a time.

- Fuzzy Side Information Clustering-Based Framework for Effective Recommendations, Wasid and Ali, 2019. Here, incorporating side-information increases the complexity of the clustering model too much, so they instead represent this side-information with fuzzy sets, to make the algorithm more efficient. The side-information is essentially just extra features in this case, so we're not dealing with constraints or cluster labels.

### Significance

The generality of the model is the strongest selling point, demonstrated in its ability to incorporate many different and flexible types of side-information. However, the significance of the paper would be improved if the authors could provide more salient baselines and comparisons in the tasks to show that the motivation for their model holds up in real-world tasks (most baseline comparisons and ablations are performed on the 2 synthetic tasks). Specifically, the paper could benefit from providing more evidence that (a) it’s generality and lack of assumptions do not significantly hurt it when stronger assumptions may be made and (b) that the incorporation of side-information in this way can actually lead to less ambiguous clusters than models which cannot incorporate such side-information.

### Pros / Cons:

* (+) Paper is well-written
* (+) The main contribution, DGC, appears to be novel
* (+) DGC is well motivated and technically principled
* (+) Interesting set of experiments, with a variety of tasks
* (-) Some relevant work not cited
* (-) Meaningful baselines only provided for some of the tasks

---

> ### Author Response · Authors · 2020-11-18
> **Thank you for you comments!**
>
> Thank you for your positive and valuable comments, suggestions, and concerns. We believe we have addressed your main concerns, including adding the references you listed and providing additional baselines to the SVHN and the CBCS experiments. We additionally address your comments and concerns in the order as they were posed in more details
>
> ### Response to the Quality Section:
>
> (1) We really appreciate and agree with your assessment. We especially agree that to demonstrate the efficacy of the proposed DGC and to corroborate the fact that it learns less ambiguous/more meaningful clusters on the CBCS dataset, a comparison/analysis with the VaDE baseline would be highly valuable. We have thus added the VaDE baseline to both the SVHN experiment and the CBCS experiment (see the Common Concern #2 in the overall comment). DGC in both cases outperformed the VaDE baseline. Please refer to the updated version of the paper for more details.
>
> (2) Please note that we use a different notation than in Kingma et. al (please see the  Clarification on DGC vs Semi-Supervised VAE (M2) in the overall comment). More specifically, the variable y in Kingma et. al, 2014 is equivalent to our cluster index variable c. The goal of this work is to discover clusters using side-information, as we assume that we never have access to the true cluster labels c. However, point (2) of the Clarification on DGC vs Semi-Supervised VAE (M2) demonstrates that Kingma et. al, 2014 is equivalent to the unsupervised portion of DGC when the cluster labels (variable y Kingma et. al, 2014, variable c in DGC) are unobserved and when the prior on z is the isotropic Gaussian distribution. We have thus added the baseline described in point (2), i.e. Kingma et. al but assuming the labels are all missing and with the augmented task network, to the SVHN experiment section as an additional comparison. DGC outperformed this baseline by a large margin both in terms of the classification accuracy and the clustering accuracy.
>
> ### Response to the Clarity Section:
> We apologize for the inadequate description of the experimental setup for the SVHN experiment---we described them in the Appendix due to the page length constraint. Now that we have an extra page for the rebuttal, we have moved some of the descriptions for the experimental setup to the main text for clarity and to help with the interpretability of the results.
>
> ### Response to the Originality Section:
> Thank you for pointing out those three works and your for your summary of them.We were not aware of these works when we wrote our paper. We have now included and discussed these references in the related work section in the updated version of our manuscript.
>
> ### Response to the Significance Section:
> Thank you for your insightful comments. We have added additional baselines (the VaDE and augmented Kingma et. al baselines) to the SVHN experiment and the VaDE baseline to the CBCS experiment as mentioned in the responses to the previous sections. We hope those added baseline comparisons will further corroborate the significance of our work.

---

### Author Response · Authors · 2020-11-18
**Overall comments and updated paper coming in a day**

We thank all four reviewers for their insightful and helpful comments. Since some of the concerns are shared among several reviewers, we briefly summarize the concerns and our solutions/explanations here. The updated version of the paper that addresses all the concerns and questions in the reviews will be uploaded at the latest by the end of next day.

### Common Concern #1: The notations are inconsistent between the left-hand side and the right-hand side in Equations (5), (7), and (8).

Our Fix:  We have addressed this inconsistency in the updated manuscript. Specifically, to make the dependencies clear, we now start with a slightly different assumption on the joint variational posterior distribution. We no longer make the mean-field assumption, but instead write: q(z,c|x,y) = q(c|z,x,y)q(z|x,y). However, note that when we compute q(z|x,y), we do not make use of the side-information y (as during test time we will not have access to y). q(c|z,x,y) is still computed using the entropy trick introduced in Sec. 4.4 when the data is unlabeled during test time. This change does not impact the methodology and implementation in our manuscript. Hence, all experimental results remain valid with this modification, but the notation indeed becomes cleaner.

### Common Concern #2: We did not compare DGC with other methods for the CBCS experiment.

Our Fix: To ensure fair comparison and examine the benefit of the probabilistically added side-information, we now provide the analysis of the clustering result obtained from applying VaDE in the updated paper (and also in the updated Appendix). As one can see, by appropriately utilizing the side-information, the differences among clusters in terms of biological properties and survival curves are much more significant, meaningful, and less ambiguous. Moreover, the p-value for the Kaplan-Meier estimator for DGC is 0.0024, compared to the p-value of 0.073 for VaDE.

### Common Concern #3: For the experiment on the SVHN dataset (Sec. 5.3), we used the ground truth labels as the side-information.

Our Explanation: We apologize for the confusion surrounding this experiment. As we clarify in the updated version of the paper, our goal for this experiment is to investigate the impact of the hyperparameter, the number of clusters desired, on the final clustering/classification accuracies. While in general one will indeed not use ground-truth labels as side-information, we used them here to obtain a good idea of what a reasonable corresponding number of clusters in latent space should be, with the expectation that an appropriate clustering in the latent space should recover a similar number of clusters. That this reasoning holds is illustrated in Table 2, where DGC achieves the best performance when the number of clusters desired is set to 10. Furthermore, Table 2 showed that DGC is able to learn to choose an appropriate number of clusters needed even when given an excessive degree of freedom, as the clustering accuracy still maintains at an acceptable level even when the number of desired clusters is set to be much larger than 10. This is especially obvious when comparing to the new baseline we will provide in the updated version of the paper, where the performance of the k-means algorithm on the features extracted from the last layer of a classification network learned on the SVHN dataset drops dramatically with the number of clusters desired increases.


### Clarification on DGC vs Semi-Supervised VAE (M2):
Multiple reviewers mentioned this comparison in their reviews, so we think it’d be best to convey our general understanding here and answer more specific questions in the individual responses.
(1) If we view semi-supervised VAE (M2) by Kingma et. al through the lens of clustering, we want to clarify that the variable y in Kingma et. al (2014) is equivalent to our clustering index variable c. Moreover, as the reviewers astutely noted, viewing  semi-supervised VAE (M2) through the lens of clustering would necessarily require the “cluster assumption”, stating that there exists a one-to-one relationship between the classes and clusters.
(2) If we assume that the prior distribution on the latent code z is an isotropic Gaussian distribution (instead of the Gaussian Mixture Model) and we use a neural net to parametrize the variational posterior distribution p(c|z,x,y) (see Common Concern #1 for the notational change), the unsupervised portion of DGC is equivalent to semi-supervised VAE (M2) when all class labels are missing. Since semi-supervised VAE (M2) is not supposed to perform well in a purely unsupervised setting, adding the task network to semi-supervised VAE (M2) serves as an augmentation. This has been added as a baseline to the SVHN experiment for additional comparison, where one can see that DGC outperforms this baseline by a large margin.

We additionally address each reviewer’s comments, concerns, and questions in the following responses.

---

### Author Response · Authors · 2020-11-19
**Just Updated Main Manuscript and Appendix**

We want to thank all reviewers again for their time and insightful comments. Due to those constructive suggestions, we have made the following changes to the updated manuscript

### Change 1
We have added the VaDE baseline to the CBCS experiment, illustrating DGC’s ability to derive less ambiguous and clinically more significant clusters by utilizing the side information

### Change 2
We have added three baselines to the SVHN experiment (results in Table 2 in the updated paper):

1. The VaDE baseline

2. The augmented semi-supervised VAE (M2) (see point (2) of the Clarification on DGC vs Semi-Supervised VAE (M2) in the other overall comment)

3. The K-means baseline, i.e. we perform K-means clustering on the features obtained from the last hidden layer of a classification network trained on the SVHN experiment (with a classification accuracy of 95.7%)
For each baseline, we vary the number of clusters desired (the hyperparameter that we want to investigate in this experiment)

### Change 3
We have cited and added more references to the related work section that were pointed out by the reviewers

### Change 4
We have clarified the notation inconsistencies that exist in Equations (5), (7), and (8)

### Change 5
We have added a pseudocode of DGC to the Appendix (the first section in the Appendix)

### Change 6
We have made some clarifications to the notations in Sec. 4.4 and have added some further explanations to clarify how DGC operates during test time when the side-information y is not available

### Change 7
We clairy the parametric modeling choices of p(y|z,c) for each experiment

### Change 8
We added the clustering result on the pacman data from hierarchical clustering with single linkage to the Appendix


Please do not hesitate to ask any questions. Thank again!

---

### Decision · Program_Chairs · 2021-01-11
**Final Decision**

**Decision:**

Reject

**Comment:**

This paper proposes a new clustering method that takes into account side information.  The paper was reviewed by four expert reviewers who expressed concerns for novelty, empirical and theoretical depth, and unclear parts of the paper. The authors are encouraged to continue research, taking into consideration the detailed comments provided by the reviewers.